# Nucleic Acid-Based Approaches to Tackle KRAS Mutant Cancers

**DOI:** 10.3390/ijms242316933

**Published:** 2023-11-29

**Authors:** Jimi Kim

**Affiliations:** 1Department of Life Sciences, Gachon University, Seongnam 13120, Republic of Korea; zimic@gachon.ac.kr; 2Department of Health Science and Technology, GAIHST, Lee Gil Ya Cancer and Diabetes Institute, Incheon 21999, Republic of Korea

**Keywords:** oncogenic KRAS, cancer therapy, RNA therapeutics, RNAi, CRISPR-Cas, mRNA vaccine

## Abstract

Activating mutations in KRAS are highly relevant to various cancers, driving persistent efforts toward the development of drugs that can effectively inhibit KRAS activity. Previously, KRAS was considered ‘undruggable’; however, the recent advances in our understanding of RNA and nucleic acid chemistry and delivery formulations have sparked a paradigm shift in the approach to KRAS inhibition. We are currently witnessing a large wave of next-generation drugs for KRAS mutant cancers—nucleic acid-based therapeutics. In this review, we discuss the current progress in targeting KRAS mutant tumors and outline significant developments in nucleic acid-based strategies. We delve into their mechanisms of action, address existing challenges, and offer insights into the current clinical trial status of these approaches. We aim to provide a thorough understanding of the potential of nucleic acid-based strategies in the field of KRAS mutant cancer therapeutics.

## 1. Introduction to Oncogenic KRAS

KRAS, a relevant proto-oncogene, was initially discovered in the early 1980s as a transforming human DNA fragment that is homologous to Kirsten sarcoma viruses (Kirsten ras or v-Kras). It is frequently mutated, with these mutations commonly implicated in the pathogenesis of pancreatic cancers, colorectal cancers (CRC), and lung adenocarcinoma [1].

KRAS is a 21 kDa small guanine triphosphatase (GTPase) that plays a pivotal role in cellular pathways governing cell survival, proliferation, and differentiation. It oscillates between the guanine diphosphate (GDP)-bound ‘OFF’ state and the guanine triphosphate (GTP)-bound ‘ON’ state. In its ‘ON’ state, KRAS exhibits enhanced affinity for multiple effectors including Raf, phosphatidylinositol 3-kinase (PI3K), and Ral guanine nucleotide dissociation stimulator (RalGDS), thus activating downstream signaling [2,3,4,5].

The biological function of KRAS is carried by the G domain, which is comprised of residues 1–166. The G domain harbors a phosphate-binding loop (p loop), and two switch regions (Switch I and II) that interact with the nucleotide. The release of the γ-phosphate through hydrolysis induces a conformation change in the two switch regions, resulting in a GDP-specific form. In addition to the G domain, KRAS has a flexible C-terminal domain, known as the hypervariable region (HVR), which is critical for its membrane localization (Figure 1a,b) [6,7].

Oncogenic mutations in KRAS primarily occur at residues G12, G13, and Q61. These mutations impair the hydrolysis of the γ-phosphate of GTP to GDP, causing KRAS to remain persistently ‘ON’, thereby constitutively activating downstream signaling and driving cellular transformation (Figure 1c,d) [8,9].

Based on the comprehensive understanding of the molecular requirements for KRAS activity, various attempts have been made in the past few decades to intercept oncogenic KRAS activity. Nevertheless, the direct inhibition of KRAS has been a significant challenge due to the picomolar affinity of the KRAS protein for guanine nucleotides, the abundance of GDP and GTP in the micromolar range, and the absence of favorable binding sites for small molecules on the surface of the KRAS protein [8,10]. Consequently, a variety of strategies have been developed to target KRAS downstream signaling pathways associated with KRAS.

In recent years, a breakthrough in developing a direct inhibitor to target mutant KRAS G12C and G12D has been made, reviving hope for the specific targeting of KRAS mutant cancers. Despite this progress, other mutants beyond G12C and G12D remain elusive to drug development. The nucleic acid-based approach may offer a promising alternative as it enables the design of inhibitors specific to any KRAS variants using only sequence information. Furthermore, it holds the potential for permanently correcting the mutations, leading to durable therapeutic outcomes.

In this review, we briefly present recent developments in small-molecule therapeutics and extensively explore the potential of versatile nucleic acid-based therapies, detailing their mechanisms, addressing current challenges, and providing updates on the clinical trial landscape of these approaches.

## 2. Targeting Mutant KRAS with Small Molecules

Small molecules directly interfering with the KRAS G12C function have been developed. Currently, two compounds, sotorasib (known as AMG510 by Amgen) and adagrasib (known as MRTX849 by Mirati Pharmaceutics), have been approved by the FDA [11,12,13,14]. Preclinical and clinical research on these approved KRAS G12C inhibitors have been extensively covered elsewhere [15,16]. Additionally, several other small molecules are under clinical trial. As of September 2023, eleven different direct inhibitors targeting KRAS G12C are under clinical investigation, as summarized in Table 1. Notably, phase 1 studies on D1553 (garsorasib) and GDC6034 (divarasib) have been published; D1553 and GDC6034 were administered to 79 patients with KRAS G12C-mutated non-small cell lung cancers (NSCLCs) and 137 patients (60 with NSCLCs, 55 with colorectal cancer, and 22 with other solid tumors), respectively. Treatment with both inhibitors resulted in encouraging antitumor activity with mostly low-grade adverse events [17,18].

Moreover, multiple drugs targeting another KRAS mutation, KRAS G12D, are in or approaching clinical trials [28]. Despite the challenging nature of G12D due to its inaccessibility to covalent chemistry, Mirati Therapeutics has developed a selective inhibitor, MRTX1133, with high affinity to G12D [29,30]. Currently, MRTX1133 is undergoing phase 1 clinical trials for patients with advanced solid tumors harboring KRAS G12D mutations (NCT05737706).

## 3. Emerging Approaches to Tackle KRAS Mutant Tumors: Nucleic Acid-Based Therapies

Nucleic acid therapeutics can be tailored based solely on gene sequence information, which allows us to address previously intractable targets with conventional modalities, such as small molecules. They selectively modulate gene expression by interacting with the transcriptome and hold promise for precise genome editing. Moreover, they can deliver the transcripts encoding tumor-associated or tumor-specific antigens, thereby facilitating the advancement of cancer immunotherapy. Thus, nucleic acid-based therapies have the potential to substantially expand the scope of targets and clinical applications in the field of medicine (Figure 2) [31,32,33,34].

### 3.1. Small Interfering RNA (siRNA)-Based Approach

#### 3.1.1. siRNA as a Therapeutic Modality

siRNAs are short, double-stranded RNAs (dsRNAs), typically 20 to 24 nucleotides in length. They directly and complementarily bind to target mRNAs, leading to target mRNA degradation. Mechanistically, siRNAs are incorporated into an Argonaute (AGO) protein, a critical component of the RNA-induced silencing complex (RISC). This process results in the unwinding of siRNA duplexes, where one strand is extracted from the duplex, while the other strand remains in the AGO to direct gene silencing in a sequence-specific manner. In this sense, siRNA therapy has the potential to target a wide range of disease-related genes [35].

Given their sequence-specific inhibition of target proteins, independently of spatial conformation, the therapeutic potential of siRNAs specifically targeting mutant KRAS has been extensively examined in both in vitro and in vivo settings over the past two decades. These siRNAs have demonstrated a selective reduction in the growth of tumors that are dependent on KRAS mutations [36,37,38,39,40]; however, their clinical translation has been hindered by intracellular and extracellular barriers. Naked and unmodified siRNAs are vulnerable to nucleases in the circulatory system, show poor pharmacokinetics, and can inadvertently silence non-targets due to their tolerance for a few base-pair mismatches [41]. In recent years, we have seen the development of various chemical modifications to increase stability, maximize potency, and minimize off-target effects, as detailed in Section 4. Additionally, delivery formulations using lipids, polymers, and exosomes have been investigated [41,42]. Currently, two siRNA-based drugs for KRAS mutant tumors—siG12D-LODER and siG12D-loaded iExosomes—are undergoing clinical trials.

#### 3.1.2. siG12D KRAS-LODER

In 2013, the Eithan Galun group, in collaboration with Silenseed Ltd., introduced a delivery platform based on a miniature biodegradable polymeric matrix for the controlled delivery of siRNA (named Local Drug EluteR, LODER). When injected into pancreatic tumor masses with the KRAS G12V mutation, the LODER-encapsulating siG12D (siG12D-LODER) significantly inhibits tumor growth in vivo [43].

In a phase 1/2a clinical trial, 15 patients with locally advanced pancreatic adenocarcinoma (PDAC) were treated with siG12D-LODER, delivered directly into the tumor. The treatment was well tolerated, with most patients showing stable disease [44]. Currently, siG12D-LODER is in a phase 2 clinical trial for patients with PDAC, administered in 12-week cycles in combination with cytotoxic chemotherapy (NCT01676259).

#### 3.1.3. iExosomes Containing siG12D

Exosomes are nano-sized extracellular vesicles (40–150 nm) enclosed within a lipid bilayer membrane, secreted by most cells and serving as crucial mediators of intracellular communication [45,46]. Kalluri and colleagues explored engineered exosomes (iEoxomes), derived from normal fibroblast-like mesenchymal cells, as efficient carriers of KRAS siG12D. iExosomes demonstrated an enhanced ability to target mutant KRAS and resulted in significant tumor reduction in vivo, attributed to CD47-dependent suppression of clearance from circulation and oncogenic KRAS-dependent enhancement of micropinocytosis [46].

A phase 1 clinical trial commenced in 2020 for PDAC patients. Approximately 28 patients received iExosomes with siG12D through intravenous injection on days 1, 4, and 10. The treatment regimen was repeated every 14 days, with up to three courses (NCT03608631).

#### 3.1.4. Recent Preclinical Advances in siRNA Therapies for Targeting KRAS Mutant Cancers

In 2023, the Alonso group conducted a study on a pan anti-KRAS siRNA with functionalized lipid nanoparticles (LNPs) [40]. LNPs are clinically proven and the most widely used carriers for delivering oligonucleotide drugs. Composed of four components—ionizable cationic lipids, cholesterol, phospholipids, and polyethylene glycol (PEG)-modified lipids—LNPs form electrostatic bonds with negatively charged nucleic acids [34,47]. However, one major limitation of LNP delivery is their tendency to accumulate in the liver [48]. To improve the intratumoral accumulation of siRNA-loaded LNPs, LNPs were functionalized with the peptide tLyp-1 (truncated Lyp-1) to target pancreatic cancer cells. When injected intravenously into mice bearing pancreatic tumors, the treatment using LNPs in combination with gemcitabine resulted in reduced tumor size and the downregulation of KRAS and phospho-ERK [40]. This highlights that the application of LNPs is not limited to liver-associated diseases and immunotherapy but is viable for developing siRNA-based therapies for KRAS mutant tumors [49].

Furthermore, various platforms for siKRAS delivery systems have been tested in preclinical settings, including cRGD peptide-modified bioresponsive chimeric polymersomes and aerosol inhalation nanoparticles (siKRAS@GCLPP NP) [50,51].

### 3.2. Anti-Sense Oligo (ASO)-Based Approach

#### 3.2.1. ASO as a Therapeutic Modality

Anti-sense oligos (ASOs) are short, single-stranded oligonucleotides, typically consisting of 15 to 20 nucleotides, designed to complementarily bind to specific RNA sequences. ASOs function through two primary mechanisms: (1) RNA cleavage and (2) steric blockage [52]. In the RNA cleavage mechanism, ASOs contain RNA bases flanking both sides of a central eight to ten DNA bases. These ASOs bind to the target RNA, forming DNA–RNA heteroduplexes, which serve as substrates for RNase H enzymes. Consequently, this binding leads to the degradation of the target RNA. In the steric blockage mechanism, ASOs with different modifications act by blocking target RNAs without inducing their degradation. This blockade results in translation arrest and the modulation of splicing by inhibiting the interaction between the target mRNA and ribosomes and by disrupting the recognition of the target RNA by the splicing factors, respectively [34,42,53,54].

ASOs have been explored as therapeutics for the treatment of KRAS mutant cancers; however, currently, there are no ASO therapeutics being tested in clinical trials. A major challenge with the use of ASOs lies in efficiently delivering them to cancer cells, similar to the challenges faced when using siRNA drugs. Recent advancements in nucleic acid chemistry and the development of innovative delivery systems are expected to overcome these obstacles. For further details on chemical modifications, please see Section 4 [55,56,57,58].

#### 3.2.2. AZD4785

A collaborative effort between AstraZeneca and Ionis Pharmaceuticals developed AZD4785, a 16-nucleotide antisense oligo, that is complementary to a sequence in the 3′ untranslated region (UTR) of KRAS mRNA. AZD4785 can induce RNase H-mediated degradation of both WT and mutant KRAS mRNAs. It adopts a novel chemistry, featuring 2′-4′ constrained ethyl (cEt) residues, which enhances potency and allows for cellular delivery without the need for additional formulations. It was shown that AZD4785 efficiently downregulates mutant and WT KRAS and has a selective phenotypic effect on KRAS mutant cells in vitro. Moreover, systemic delivery of unformulated AZD4785 showed antitumor activity in KRAS mutant lung cancer xenografts and patient-derived xenograft (PDX) mouse models [32]. In 2017, a phase 1 clinical trial was initiated for 28 patients with advanced solid tumors, who received intravenous injection of AZD4785; however, no further clinical trials have been conducted and no published results are available.

#### 3.2.3. Recent Preclinical Advances in ASO Therapy for Targeting KRAS Mutant Cancers

In 2022, Ke Zhang and colleagues reported a novel approach involving PEGylated oligonucleotides, termed polymer-assisted compaction of DNA (pacDNA). PacDNA consists of one to five ASOs tethered to a bottlebrush PEG structure, effectively concealing the ASOs within a PEG environment. This design minimizes non-specific ASO–protein interactions, thereby enhancing their biopharmaceutical properties. The elevated plasma concentration of pacDNA leads to the passive targeting of pacDNA to highly vascularized tissues such as tumors.

PacDNA, engineered to include ASO sequences to target the 3′UTR of KRAS mRNA, was intravenously administered to mice bearing subcutaneous NSCLC xenografts. The systemic delivery of pacDNA resulted in potent KRAS downregulation and tumor growth inhibition. Notably, the preclinical results were comparable to those achieved using AZD4785, even with merely 0.0005-times the dosage of AZD4785. PacDNA is a promising therapeutic platform for addressing KRAS-driven human cancers and warrants further exploration in both preclinical and clinical settings [31].

### 3.3. CRISPR-Cas-Based Approach

#### 3.3.1. Clustered Regularly Interspaced Short Palindromic Repeats (CRISPR)-CRISPR-Associated Endonuclease (Cas) System

The CRISPR-Cas system is a widely used genome-editing technology for mammalian cells and organisms. This system relies on the designed guide RNA (gRNA) for guidance and target specificity. The guide RNA forms a ribonucleoprotein complex with Cas nuclease and recognizes target genes positioned adjacent to a protospacer-adjacent motif (PAM) based on base pairing. Subsequently, Cas cleaves double-stranded DNA (dsDNA) or single-stranded RNA (ssRNA), leading to the irreversible knock-out or knock-in of the target gene [34,59,60].

Several in vitro studies have used CRISPR-based cleavage approaches to disrupt mutant KRAS, and have demonstrated the potential of this system for treating patients with KRAS mutant cancers [61,62,63,64,65,66]. However, the clinical application of the CRISPR-based approach has been limited by issues such as off-target mutations, toxicity, and delivery efficacy. CRISPR-Cas-mediated dsDNA breaks often trigger apoptosis and can result in the emergence of inactivating mutations in TP53. Unintentional editing of a genome can lead to profound long-term complications [67,68,69,70]. Pre-existing anti-Cas antibodies in humans may further diminish editing efficiency [71]. Delivering the CRISPR system is particularly challenging due to its large size and the inadequate accessibility to the desired targets. Viral vectors are primarily used for CRISPR/Cas-mediated gene therapy [68,72]; however, their application is mostly confined to ex vivo manipulation, and delivering them to the specific tumor tissue remains challenging. Moreover, viral vectors pose the risk of integrating into the host genome and causing persistent expression, which can potentially increase the incidence of tumorigenesis, immunogenicity, and off-target effects [72]. Non-viral vectors, such as LNP and polymers, have been explored, offering attractive alternatives [73,74]. Much work on improving targeting specificity and efficient delivery to tumor tissues will be required for the CRISPR system to emerge as a viable clinical intervention for KRAS mutant cancers. Here, we discuss the current advancements in the CRISPR-Cas system that enable the permanent correction of mutant KRAS and the disruption of RNA rather than DNA.

#### 3.3.2. Recent Preclinical Advancements in CRISPR-Cas9 Therapy Targeting KRAS Mutant Cancers

CRISPR-Cas9 base editors (BEs) have been developed to enable precise base conversion without DNA cleavage, allowing for the correction of KRAS mutations. This system depends on deaminase enzymes, facilitating the transition from C to T or A to G (Figure 3a) [75].

Buchholz and colleagues demonstrated that a Cas9 variant combined with the enhanced adenine base editor, ABE8e, can induce adenine base conversions, resulting in efficient correction of G12S/G12D mutations as well as a moderate correction of G13D mutation to the WT in both NSCLC cells and patient-derived organoids (Figure 3b). This correction led to impaired cell proliferation and organoid growth [76]. With this system, no appreciable unintended editing of potential DNA sites and transcriptomes was detected, with only weak bystander editing at a neighboring adenine, causing a silent mutation. This illustrates the robust and specific nature of ABEs [76].

Furthermore, the next-generation editing technique, termed prime editing (PE), has emerged. PE adopts a fusion protein composed of an engineered reverse transcriptase and a catalytically inactivated Cas9 nickase. The editor complex is guided by prime-editing guide RNA (pegRNA), which contains primer binding sites and a reverse transcription template for the desired correction (Figure 3c). This system induces nicks at the target sites, enabling transcription of the cleaved strand and the introduction of the intended edits [77].

In 2023, Kim and colleagues successfully utilized the PE system to correct G12 and G13 KRAS mutations using universal pegRNAs. They delivered all the components to KRAS mutant cells using lipofectamine and achieved a notable correction of endogenous KRAS mutations, with up to a 30% correction frequency in the cell lines (Figure 3d) [78]. Although the study did not investigate the functional outcomes of the genetic corrections, the findings underscore the potential of the PE system in the development of precise therapeutic strategies targeting mutant KRAS.

#### 3.3.3. RNA-Editing CRISPR/Cas13 Systems Targeting KRAS Mutant Cancers

While the Cas9 protein is the most commonly used Cas protein, cleaving target dsDNAs and activating nonhomologous end-joining repair and consequent gene alteration, Cas13 is another Cas protein that specifically recognizes and cleaves RNA instead of DNA. This makes it a potent alternative approach to controlling the expression of oncogenic transcripts in cancer cells [60,65,67,79,80].

It has been shown that the CRISPR-Cas13a system specifically downregulates KRAS G12D mRNA without disrupting WT mRNA, and shows significant antitumor activity not only in vitro but also in vivo [65]. In an in vivo study, the CRISPR-Cas13a system targeting KRAS G12D was injected intratumorally into mice bearing pancreatic tumor xenografts. Despite being a long way from entering clinical trials, the advent of these novel approaches provides hope for effectively targeting KRAS mutant tumors.

### 3.4. mRNA-Based Approach

#### 3.4.1. mRNA Vaccine

The recent success of mRNA-based vaccines for COVID-19 has fueled great hope for cancer therapeutics. Individuals’ tumors carry a unique set of somatic mutations that can be recognized by the host immune system as “non-self”, making them promising targets for cancer vaccines. In vitro-synthesized mRNAs are delivered to the cytoplasm and translated into the targeted antigens. An mRNA-based vaccination trains the immune system to activate B cell-mediated humoral immunity and CD4+/CD8+ T cell-mediated immunity against cancer-specific antigens [81,82].

mRNA therapeutics enable the cost-effective and rapid design and production of any functional antigens within the human body, offering a novel path for personalized cancer vaccines [83]. The concept of personalized cancer vaccines was first explored in patients with melanoma. Among the non-synonymous mutations identified in these patients through next-generation sequencing, ten selected mutations were used to generate mRNA vaccines [84,85]. However, the challenges of neoantigen vaccines include identifying the most effective antigen or combination of antigens with high immunogenicity and enabling the immune system to penetrate deep into tumors [86].

#### 3.4.2. mRNA-5671

In 2018, Moderna and Merck collaborated to develop an mRNA vaccine against the most prevalent KRAS mutations (KRAS G12C, G12D, and G12V), known as mRNA-5671(V941). This mRNA expressing the KRAS epitope is encapsulated in LNP and administered via the intramuscular route every 3 weeks for a total of nine cycles [82,87]. A phase 1 clinical trial involving monotherapy and in combination with a PD-1 inhibitor, pembrolizumab, has been completed with a cohort of 70 patients with KRAS mutant advanced or metastatic NSCLC, CRC, and PDAC (NCT03948763); however, as of now, no results from this trial have been published.

#### 3.4.3. Circular RNA Vaccine

Circular RNAs (circRNAs) are a recently emerged class of RNAs, forming covalently closed structures produced by the back-splicing of precursor mRNA. They exhibit increased stability compared to linear mRNAs due to their closed ring structure, which protects them from exonuclease-mediated degradation [88,89]. Engineered circRNAs have demonstrated promising anti-tumor effects by expressing an immunogen or a mixture of cytokines that induce an immune response against tumors [90,91]. These studies suggest that circRNAs hold great potential to become an effective platform for RNA-based therapeutics targeting KRAS mutant tumors.

### 3.5. Immunomodulating Oligonucleotides

Vaccines, including mRNA-based vaccines, provide a new opportunity for the treatment of KRAS mutant cancers [10]. Cancer vaccines have gained recognition as a promising therapeutic strategy that triggers a robust tumor-specific immune response, subsequently suppressing tumor growth. However, the effectiveness of these cancer vaccines can be hindered by the oncogenic KRAS-induced reprogramming of the tumor microenvironment, leading to cancer cell immune evasion [92,93,94,95]. To boost the immunogenicity of a cancer vaccine, adjuvants can be employed. Toll-like receptor (TLR) agonists are increasingly being used as a cancer vaccine adjuvant [96,97].

#### ELI-002

Elicio Therapeutics has developed a novel immunotherapy, ELI-002, comprising a lymph-node-targeted amphiphile (Amph)-modified oligonucleotide (Amph-CpG-7909) and a mixture of Amph-conjugated peptide-based antigens that target mutant KRAS [98]. CpG-7909 is a synthetic oligonucleotide designed to specifically agonize TLR9, resulting in the stimulation of B cell proliferation, increased production of antigen-specific antibodies, and the induction of interferon alpha production and interleukin-10 secretion [99]. In a phase 1 trial, ELI-002 was administered via subcutaneous injection as an adjuvant treatment of minimal residual disease in 22 patients with KRAS/NRAS mutant pancreatic cancer (NCT05726864). The trial demonstrated no dose-limiting toxicities, with all grade 1 adverse events [98]. Phase 1/2 trials are currently ongoing for patients with KRAS/NRAS mutant solid tumors (NCT04853017).

## 4. Chemical Modifications for Nucleic Acid-Based Drugs

Nucleic acid-based drugs must overcome multiple challenges to be effective: they must evade serum nucleases, avoid uptake by scavenger macrophages, reach the targeted tissue, and enter the desired cells via endocytosis without triggering a detrimental immune response. To enhance their delivery, researchers have explored a range of strategies, such as chemical modification, the direct conjugation of ligands to oligonucleotides for tissue-specific accumulation, and encapsulation within various nanocarriers [42,49,100].

### 4.1. Chemical Modification of siRNA and ASO

The phosphorothioate backbone (PS) and 2′-O-methyl (2′-O-Me) group on the ribose sugar are the most common modifications applied to small oligonucleotide drugs, such as ASOs and siRNAs (Figure 4a,b). These modifications are designed to increase the drug’s stability, cellular uptake, bioavailability, and target RNA-binding affinity [100,101,102]. Notably, ASOs targeting mutant KRAS have been engineered with PS modifications and 2′-4′ constrained ethyl (cEt) residues instead of 2′-O-Me to further augment their effectiveness [32,103]. cEt is a methylated derivative of locked nucleic acid (LNA). Other sugar modifications widely employed in the RNA interference (RNAi) system include 2-fluoro (2′-F) and 2′-O-methylethyl (2′-O-ME), which enhance the binding affinity and provide resistance to enzymatic degradation (Figure 4b) [17,100].

The significant impact of these chemical modifications on the pharmacokinetics and overall efficacy of oligonucleotide therapies has driven rapid progress in this field, as extensively reviewed in the literature [41,104,105,106].

### 4.2. Chemical Modification of mRNA

Exogenously administered mRNAs often trigger immune responses by activating pattern recognition receptors, leading to suppressed translation [107,108]. To reduce immunogenicity, naturally occurring modified nucleosides can be incorporated into mRNAs during in vitro transcription, such as pseudouridine (Ψ), N1-methylpseudouridine (m1Ψ), 5-methyluridine (m5U), 5-methylcytidine (m5C), and N6-methyladenosine (Figure 4c) [109,110]. These modifications promote increased stability and efficient translation [111].

## 5. Conclusions and Future Perspectives

Significant progress has recently been made in the pursuit of targeting oncogenic KRAS. Particularly, nucleic acid-based strategies have shown promise as potential alternatives to conventional therapies. An increasing number of nucleic acid-based therapeutics targeting oncogenic KRAS have undergone clinical trials, as summarized in Table 2. Previously, the application of RNA as a therapeutic molecule was limited by its short-lasting property in circulation due to enzymatic degradation and rapid renal clearance. In addition, challenges also lay in the inefficient intracellular delivery of RNA to the desired tissue. However, the past decade has witnessed the development and application of various chemical modifications to the nucleic acid backbone, sugars, and nucleobases, resulting in improved chemical and biological stability, enhanced specificity, and better delivery. Furthermore, the development of versatile delivery systems, such as exosomes, LNP, and polymer-based formulations, has propelled nucleic acid-based therapeutics closer to clinical application. Notably, recent advances in the CRISPR-Cas system have opened new and highly specific avenues for inhibiting KRAS mutant cancers. The optimization of delivery platforms to maximize the ability to target cancer cells will be necessary to make the system more powerful. What sets nucleic acid-based therapeutics apart is their ability to rapidly design and produce functional targets, enabling personalized treatment. Addressing acquired resistance in KRAS-driven tumors, which can emerge after treatment with KRAS-specific inhibitors, is another promising avenue for harnessing nucleic acid-based therapeutics.

## Figures and Tables

**Figure 1 ijms-24-16933-f001:**
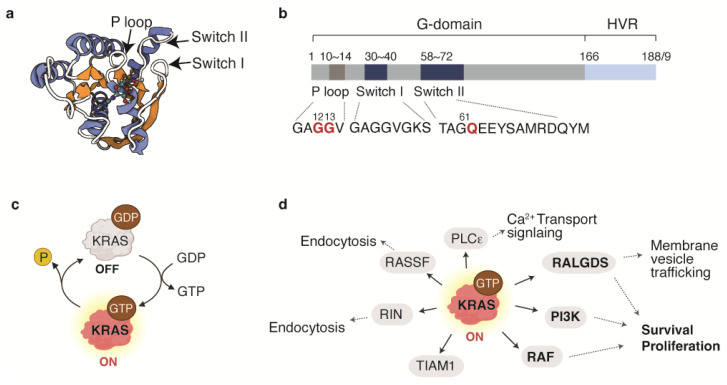
Structure and function of KRAS. (**a**) Crystal structure of GDP-bound KRAS (PDB ID: 4ebo, created with BioRender.com). (**b**) Domain structure of KRAS, consisting of the G domain and hypervariable region. Within the G domain, three regions—P loop, switch I, and switch II—are responsible for binding guanine nucleotides and activating downstream signaling. (**c**) The function of KRAS as a binary switch. (**d**) Key downstream effectors that mediate KRAS signaling and the cellular impact of KRAS activation.

**Figure 2 ijms-24-16933-f002:**
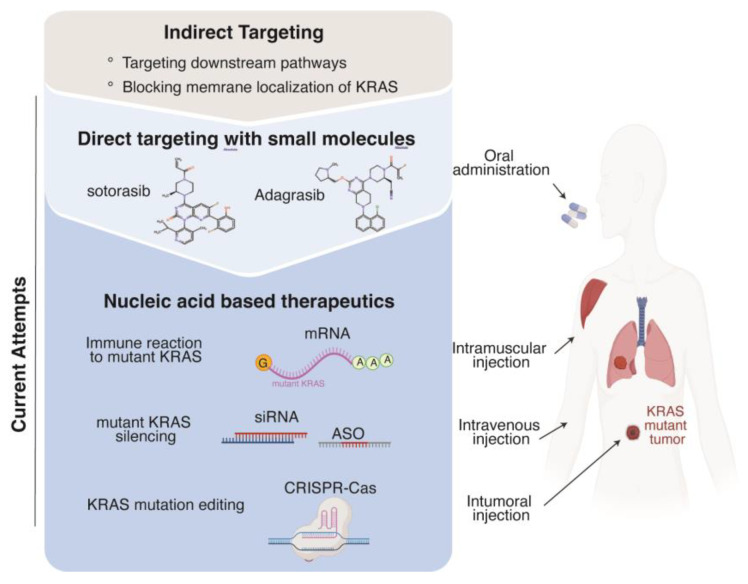
Strategies for targeting KRAS mutant cancers. Earlier strategies for targeting KRAS mutant cancers involve targeting the downstream signaling pathways and blocking the membrane localization of KRAS. Recently, several KRAS-specific inhibitors have been successfully developed. Moreover, nucleic acid-based approaches have shown promise in preclinical studies and are currently under investigation in clinical trials.

**Figure 3 ijms-24-16933-f003:**
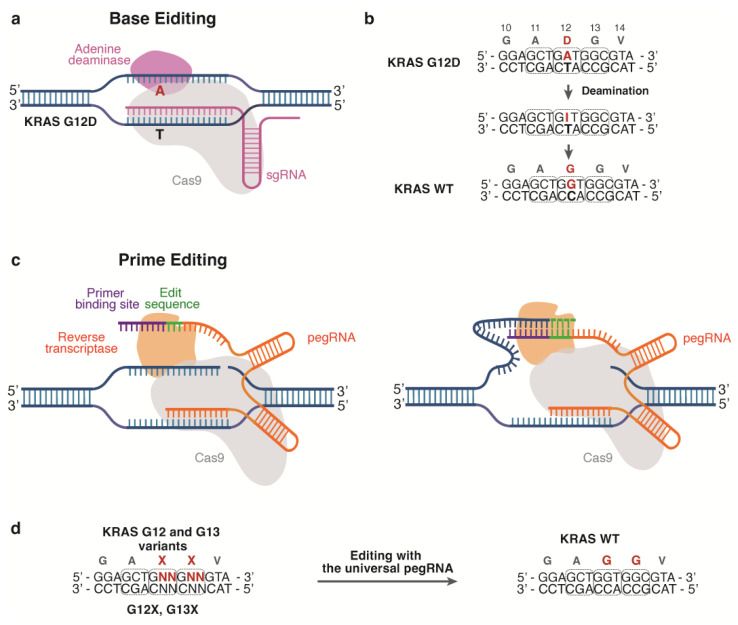
CRISPR-Cas-based approaches to edit KRAS mutations. (**a**) Schematic diagram of base editing with an adenine base editor (created with BioRender.com). (**b**) Repair of the KRAS G12D mutation by employing sgRNA. Adenosine deamination converts adenosine to inosine, which has a base-pairing preference with guanosine during DNA replication. (**c**) Schematic diagram of prime editing. Prime editors employ an engineered reverse transcriptase and a catalytically inactivated Cas9 nickase and pegRNA (created with BioRender.com). (**d**) KRAS correction with universal pegRNAs. The pegRNA contains a sequence that spells the desired sequence change, allowing the correction of KRAS mutations.

**Figure 4 ijms-24-16933-f004:**
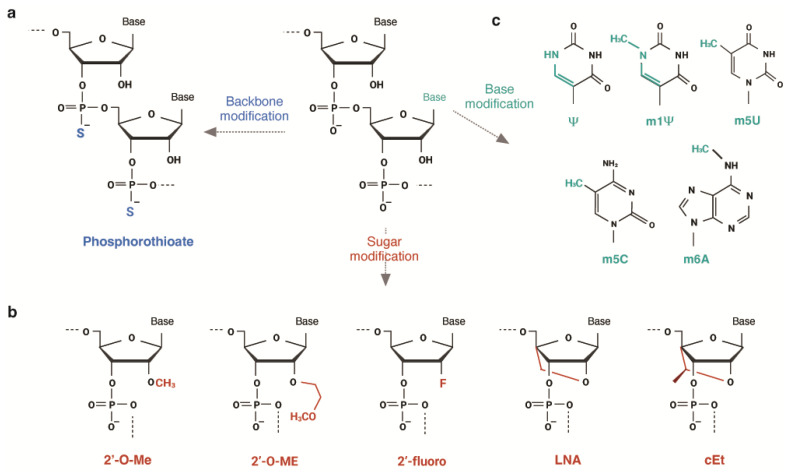
Chemical modifications for nucleic acid-based drugs. (**a**) Phosphorothioate modification: one of the non-bridging oxygen atoms is replaced with sulfur. (**b**) Sugar modifications: 2′-O-Me (2′-O-methyl group), 2-O-ME (2′-O-methylehtyl group), 2′-fluoro, LNA(locked nucleic acid), and cEt (constrained ethyl group). (**c**) Base modifications: Ψ (pseudouridine), m1Ψ (N1-methylpseudouridine), m5U (5-methyluridine), m5C (5-methylcytidine), and m6A (N6-methyladenosine).

**Table 1 ijms-24-16933-t001:** Ongoing clinical trials for small molecules directly targeting KRAS G12C.

Drug	States	Sponsor	Reference
BI1823911	Phase 1 (NCT04973163)	Boehringer Ingelheim (Ingelheim, Germany)	[19]
BPI-421286	Phase 1 (NCT05315180)	Betta Pharmaceuticals Co., Ltd. (Hangzhou, China)	[20]
D-1553	Phase 1/2 (NCT04585035, NCT05492045)	InventisBio Co., Ltd. (Shanghai China)	[17,21]
D3S-001	Phase 1 (NCT05410145)	D3 Bio Co., Ltd. (Wuxi, China)	[22]
GDC-6036	Phase 1/2 (NCT04449874, NCT05789082)	Genentech, Inc. (San Francisco, CA, USA)	[18,23,24]
GEC255	Phase 1 (NCT05768321)	GenEros Biopharma Hangzhou Ltd. (Hangzhou, China)	
GFH925	Phase 1 (NCT05005234) Phase 1b/3 (NCT05504278)	Innovent Biologics Co., Ltd. (Suzhou, China)	
HS-10370	Phase 1/2 (NCT05367778)	Jiangsu Hansoh Pharmaceutical Co., Ltd. (Lianyuangang, China)	
JAB-21822	Phase 1/2 (NCT05002270, NCT05009329, NCT05194995, NCT05276726, NCT05288205)	Jacobio Pharmaceuticals Co., Ltd. (Beijing, China)	[25]
JDQ443	Phase 1/2 (NCT04699188, NCT05358249) Phase 2 (NCT05445843) Phase 3 (NCT05132075)	Novartis Pharmaceuticals (Basel, Switzerland)	[26,27]
MK1084	Phase 1 (NCT05067283)	Merck Sharp & Dohme LLC (Rahway, NJ, USA)	
YL-15293	Phase 1/2 (NCT05119933)	Shanghai YingLi Pharmaceutical Co., Ltd. (Shanghai, China)	

**Table 2 ijms-24-16933-t002:** Clinical trials in nucleic acid-based drugs targeting KRAS mutant tumors.

Class	Drugs	States	Sponsor	Reference
siRNA	siG12D KRAS_LODER	Phase 1 (NCT01188785) Phase 2 (NCT01676259)	Silenseed Ltd. (Jerusalem, Israel)	[44,45]
	3 iExosomes containing siG12D	Phase 1 (NCT03608631)	AstraZeneca (Cambridge, England) Ionis Pharmaceuticals (Carlsbad, CA, USA)	[47]
RNA vaccine	mRNA-5671	Phase 1 (NCT03948763)	Moderna (Cambridge, MA, USA) Merck & Co., Inc. (Rahway, NJ, USA)	[79,84]
	RNA tumor vaccine	Phase 1 (NCT05202561)	Hospital Bengbu Medical College (Bengbu, China)	
Immunomodulating oligonucleotides	ELI-002	Phase 1/2 (NCT04853017) (NCT05726864)	Elicio Therapeutics (Boston, MA, USA)	[91,92]

## Data Availability

Not applicable.

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
