# Peer review of "Nucleic Acid-Based Approaches to Tackle KRAS Mutant Cancers"

_ijms, 2023, doi:10.3390/ijms242316933_

Round 1

Reviewer 1 Report

Comments and Suggestions for Authors

The manuscript focused on the comprehensive review of nucleic acids-based therapeutics targeting oncogenic KRAS. The figures are illustrative and easy to understand to the readership in the journal. The exciting and emerging application of the genome editing technology in this field is also discussed and reviewed in the manuscript. However, there are several major recommendations that must be addressed before the final acceptance of the manuscript. 

1. The discussion of small molecules targeting KRAS mutation appears somewhat tangible to the main focus of the manuscript. And the connection between the small molecule drugs and the nucleic acid-based therapeutics remains unclear. It is highly recommended to provide a more thorough examination of this topic, elucidating its relevance to nucleic acid-based therapeutics. Additionally, it would be beneficial to explore whether nucleic acid-based therapeutics can address any challenges associated with small molecule drug targeting of KRAS mutations.

2. The discussion of siRNA-based interference in the manuscript is currently quite general. In addition to the existing content, the advantages of using siRNA-based therapeutics, the challenges and limitations in designing delivery vehicles, targeting ligands and some aspects regarding pharmacology to advance the siRNA-based therapeutics should be provided. 

3. CRISPR/Cas9 genome editing tool is emerging in targeting oncogenic mutation. While the author acknowledged the limitations of double-stranded DNA breaks and off-target efficiency in the manuscript, it is essential to emphasize two major challenges in the CRISPR/Cas9-based therapy: targeted delivery efficiency and the associated potential immunogenicity. Given that, the direct application of CRISPR/Cas9 in the anti-tumor therapy may be less promising than siRNA and small molecule-based therapy. The significance of CRISPR/Cas9 technology in targeting oncogenic KRAS mutation, the design of delivery vehicle, the correlation between editing efficiency and final anti-tumor efficacy is recommended to be included for the comprehension of the section. 

4.  In addition to mRNA vaccine, circular RNA vaccine is also emerging as RNA-based approach to in anti-tumor vaccine that can revolutionize the nucleic acid-based therapeutics targeting KRAS mutations. The discussion regarding circular RNA is recommended to be provided. 

5. Chemical modifications of RNA play a crucial role in the success of RNA-related therapeutics. To enhance the overall flow of the manuscript and emphasize the significance of these modifications in improving therapeutic outcomes, it is advisable to integrate discussions about chemical modifications into each relevant section of the manuscript.

Comments on the Quality of English Language

The manuscript presents a good quality of English language. 

Author Response

We would like to thank the reviewers for the comprehensive and constructive review of our manuscript. Following the suggestions, we have carefully revised our manuscript. We believe that the updated version now addresses all the concerns previously raised.

Please find a point-by point description of the changes and additions made in response to the reviewers’ comments.

Reviewer 2 Report

Comments and Suggestions for Authors

The author, Kim J, in his manuscript "Nucleic Acid-Based Approaches to Tackle KRAS mutant cancers" presents different novel types of therapies for tumors containing KRAS mutations.  After short introduction, clinical trials with small inhibitors targeting specific KRAS mutations are described. The main part of the review describes novel nucleic acid based approaches, from different antisense oligonucleotides to CRISPR-Cas approach. Furthermore, immunomodulating oligonucleotides are presented, as well as types of chemical modifications of the therapeutic molecules for avoiding degradation in the serum and enhancing delivery.

The manuscript is well written, with data and references up to date, in logic order. Although describing complex material it is very comprehensible and illustrated with figures additionally presenting the principles of each therapy.

Minor comments

Clinical trials using nucleic acid based therapies could also be presented in the form of table.

Figure 3: a Base Editing

The main question addressed by the manuscript of Kim J "Nucleic acid-based approaches to tackle KRAS mutant cancers" is the nucleic acid based therapy against KRAS mutated cancers. The topic is relevant in the field as it describes novel therapies in systematic and comprehensive way. Although there is a huge number of reviews considering KRAS therapies, they are mostly oriented on small inhibitors or on specific types of tumor. Also, the manuscript presents some novel methods and trials which have started recently and are still going on.
Conclusions are clearly presented, together with future perspectives. References are appropriate.
Tables and figures are illustrative and adequate. I would suggest adding a table with a list of new nucleic acid based therapies and their status regarding clinical trials.

Author Response

(The authors gave the same response as above.)

Round 2

Reviewer 1 Report

Comments and Suggestions for Authors

All the major recommendations have been addressed by the authors, and the paper is in an acceptable state.

Comments on the Quality of English Language

The quality of English language is suitable for the publication.